

# Sequence and structure analyses of lytic polysaccharide monooxygenases mined from metagenomic DNA of humus samples around white-rot fungi in Cuc Phuong tropical forest, Vietnam

Nam-Hai Truong[1,2], Thi-Thu-Hong Le[1,2], Hong-Duong Nguyen[1], Hong-Thanh Nguyen[3], Trong-Khoa Dao[1], Thi-Minh-Nguyet Tran[4], Huyen-Linh Tran[1], Dinh-Trong Nguyen[1], Thi-Quy Nguyen[1], Thi-Hong-Thao Phan[1], Thi-Huyen Do[1,2], Ngoc-Han Phan[5], Thi-Cam-Nhung Ngo[5] and Van-Van Vu[5]

[1] Institute of Biotechnology (IBT), Vietnam Academy of Science and Technology (VAST), Hanoi, Vietnam
[2] Graduate University of Science and Technology (GUST), Vietnam Academy of Science and Technology (VAST), Hanoi, Vietnam
[3] Vimec Hi-tech Center, Vinmec Healthcare system, Hanoi, Vietnam
[4] The Key Laboratory of Enzyme and Protein Technology (KLEPT), VNU University of Science, Hanoi, Vietnam
[5] NTT Hi-Tech Institute, Nguyen Tat Thanh University, Ho Chi Minh, Vietnam

Corresponding authors
Nam-Hai Truong, tnhai@ibt.ac.vn
Van-Van Vu, vanvu@ntt.edu.vn

## ABSTRACT

**Background**. White-rot fungi and bacteria communities are unique ecosystems with different types of symbiotic interactions occurring during wood decomposition, such as cooperation, mutualism, nutritional competition, and antagonism. The role of chitin-active lytic polysaccharide monooxygenases (LPMOs) in these symbiotic interactions is the subject of this study.

**Method**. In this study, bioinformatics tools were used to analyze the sequence and structure of putative LPMOs mined by hidden Markov model (HMM) profiles from the bacterial metagenomic DNA database of collected humus samples around white-rot fungi in Cuc Phuong primary forest, Vietnam. Two genes encoding putative LPMOs were expressed in *E. coli* and purified for enzyme activity assay.

**Result**. Thirty-one full-length proteins annotated as putative LPMOs according to HMM profiles were confirmed by amino acid sequence comparison. The comparison results showed that although the amino acid sequences of the proteins were very different, they shared nine conserved amino acids, including two histidine and one phenylalanine that characterize the H1-Hx-Yz motif of the active site of bacterial LPMOs. Structural analysis of these proteins revealed that they are multidomain proteins with different functions. Prediction of the catalytic domain 3-D structure of these putative LPMOs using Alphafold2 showed that their spatial structures were very similar in shape, although their protein sequences were very different. The results of testing the activity of proteins GL0247266 and GL0183513 show that they are chitin-active LPMOs. Prediction of the 3-D structures of these two LPMOs using Alphafold2 showed that GL0247266 had five functional domains, while GL0183513 had four functional domains, two of which that were similar to the GbpA_2 and

GbpA_3 domains of protein GbpA of *Vibrio cholerae* bacteria. The GbpA_2 - GbpA_3 complex was also detected in 11 other proteins. Based on the structural characteristics of functional domains, it is possible to hypothesize the role of chitin-active GbpA-like LPMOs in the relationship between fungal and bacterial communities coexisting on decomposing trees in primary forests.

## INTRODUCTION

Lignocellulose and chitin are two polysaccharides that are abundant in nature and can potentially replace fossil sources in the production of fuels, chemicals, and other materials. However, one of the obstacles in the use of these alternative carbon sources to produce fuel is their recalcitrance to degradation by hydrolytic enzymes. Recently, it has been shown that new enzymes belonging to the lytic polysaccharide monooxygenase (LPMO) family are capable of catalyzing the oxidative degradation of recalcitrant polysaccharides (*Vaaje-Kolstad et al., 2010*; *Hemsworth, Davies & Walton, 2013*; *Beeson et al., 2015*). The discovery of LPMOs shows the occurrence in nature of the enzymatic metabolism of polysaccharides, especially recalcitrant materials such as chitin and lignocellulose (*Eijsink et al., 2019*).

LPMOs are copper-dependent biocatalysts that use molecular oxygen and electron donor to break glycosidic bonds (*Quinlan et al., 2011*; *Aachmann et al., 2012*; *Hemsworth et al., 2013*). LPMO crystal structures of some bacteria *Serratia marcescens*, *Enterococcus faecalis* (*Vaaje-Kolstad et al., 2012*), *Vibrio cholerae* (*Wong et al., 2012*), *Streptomyces coelicolor* (*Forsberg et al., 2014a*), *Jonesia denitrificans* (*Votvik et al., 2023*), *Bacillus amyloliquefaciens* (*Hemsworth, Davies & Walton, 2013*) and fungi *Hypocrea jecorina* (*Karkehabadi et al., 2008*), *Neurospora crassa* (*Li et al., 2012*), *Phanerochaete chrysosporium* (*Wu et al., 2013*), *Thermoascus aurantiacus* (*Quinlan et al., 2011*) were studied using both X-ray crystallography and nuclear magnetic resonance (NMR) methods. The crystal structure analysis shows that the core of the catalytic domain is a $\beta$-sandwich of seven to nine $\beta$-strands. The loops connecting these $\beta$-strands constitute the flat substrate binding surface including two conserved histidines coordinating with a single copper atom, also known as a histidine brace (*Hemsworth, Davies & Walton, 2013*; *Beeson et al., 2015*; *Vaaje-Kolstad et al., 2017*). It was noted that the catalytic domain of LPMO is capable of binding to the substrate (*Lundell et al., 2014*; *Bissaro et al., 2018a*) and variation in the substrate specificity of LPMOs is determined by the copper active site configuration (*Forsberg et al., 2014a*).

Different types of LPMOs have been found capable of hydrolyzing cellulose, chitin, starch, and cellodextrins (*Zhou & Zhu, 2020*). Some LPMOs are also active on xylan (*Tõlgo et al., 2022*). Based on sequence characteristics, LPMOs are classified according to CAZY auxiliary activity (AA) families 9-11, 13, 14, 15, and 16 (www.cazy.org). Families AA9, 11, and 13-16 are mainly from fungi, whereas family AA10 is detected in bacteria (*Zhou &*

*Zhu, 2020*). Families AA9 and AA10 are active against both chitin and cellulose (*Borisova et al., 2015*), while family AA13 is active against starch (*Vu et al., 2014*; *Lo Leggio et al., 2015*). There are three types of fungal cellulose active LPMOs: those that only oxidize C1, those that only oxidize C4, and those that oxidize both C1 and C4. Most cellulose active bacterial LPMOs known to date only oxidize C1. None of them have been shown to oxidize only C4 and some bacterial LPMOs can oxidize both C1 and C4 (LPMOs have tyrosine instead of phenylalanine, *e.g.*, *Sc* LPMO10B (PDB ID 4OY6) or TfE7 (PDB ID 4GBO). The oxidized product of chitin in chitin-active LPMO reactions has only been shown to be at C1; C4-oxidized products have not been confirmed (*Beeson et al., 2012*; *Forsberg et al., 2014a*).

Bacterial LPMOs participate in various physiological processes such as nutrition, endosymbiosis, antagonism, and host pathogenesis (*Agostoni, Hangasky & Marletta, 2017*). In endosymbiosis, the host organisms will take advantage of enzymes secreted by bacteria such as LPMO, glycosidic hydrolase, and cellulases to create new ways of feeding. For example, some insect species take advantage of the decomposing enzymes of the symbiotic *Streptomyces* sp. to facilitate energy accumulation within the plant cell wall for larval development (*Seipke, Kaltenpoth & Hutchings, 2012*). Some bacterial LPMOs have been found to have antifungal activity due to their ability to specifically bind to chitin that present in fungal cell walls (*Bowman & Free, 2006*). For example, the Cbp50 protein of the soil bacterium *Bacillus thuringiensis* has the ability to bind strongly to β-chitin in fungal cell walls, thereby preventing the synthesis of chitin during fungal cell division (*Mehmood et al., 2011*). In some cases, bacterial LPMO can act as a virulence factor. For example, Kirn and colleagues (*Kirn, Jude & Taylor, 2005*) hypothesized a role for GbpA of V. cholerae in human infections by suggesting that GbpA (as a colonization factor) promotes bacterial adhesion to intestinal mucosal surfaces. The mechanism for GbpA-mediated bacterial colonization of intestinal cells was suggested to follow this sequence: *V. cholerae* constitutively produces GbpA prior to colonization, mainly as a secreted protein that adheres to the *V. cholera* surface; once in the host intestine, GbpA attaches itself *via* domain 1 (catalytic domain) to mucin, thereby marking the surface for *V. cholerae* colonization; and domains 2–3 (GbpA_2 and GbpA_3 domains) of the protein then bind to the *V. cholerae* surface, enhancing micro-colony formation (*Wong et al., 2012*).

White-rot fungi or brown-rot fungi are the common names for wood-decaying fungi belonging to the order Basidiomycetes, class Agaricomycetes in the phylum Basidiomycota (*Lundell et al., 2014*; *Bissaro et al., 2018a*). The most widely-studied and popular white-rot fungi genera are Phanerochaete, Trametes, Pleurotus, and Ganoderma. Fungi from these genera are characterized by their ability to naturally degrade lignocellulosic materials, including lignin, hemicellulose, and cellulose (*Suryadi et al., 2022*). In nature, the biodegradation of wood occurs under various environmental conditions and is mainly caused by fungi, but also by bacteria. Both organisms can degrade wood individually as well as in combination (*Rehbein et al., 2013*). It is assumed that there is a synergy between the fungal and bacterial communities in wood decomposition (*Embacher et al., 2023*). There are two current concepts related to the role of bacteria in wood decomposition. The first is that bacteria colonize wood at the beginning of the decomposition process
and thus have a synergistic effect in creating conditions for fungi to grow on wood. The alternative hypothesis is that there was already a fungal community established in the decaying wood because the fungus was actually latently present while the tree was alive (*Parfitt et al., 2010*), and this second hypothesis is most likely valid regarding wood decay under natural conditions (*Embacher et al., 2023*). The co-occurrence of bacteria and fungi on plant-derived substrate can induce different types of symbiotic interactions such as cooperation, mutualism, nutrient competition, and antagonism (2005; *Johnston, Boddy & Weightman, 2016*). Bacteria can promote the growth of fungal mycelium (*Schulz-Bohm et al., 2017*; *Oh et al., 2018*), the formation of mycorrhizal (*Frey-Klett, Garbaye & Tarkka, 2007*; *Tarkka, Drigo & Deveau, 2018*), and production of secondary metabolites in fungi (*Vahdatzadeh, Deveau & Splivallo, 2015*; *Tauber et al., 2018*). In return, fungi provide a large number of habitats and niches for bacteria. For example, bacteria have been found to be associated with fungal hyphae, fungal spores, mycorrhizal roots, and fungal fruiting bodies; inside fungal cells (endofungal bacteria); and on the surface of fungi (ectofungal bacteria) (2005; *Bonfante & Anca, 2009*).

Hidden Markov models (HMMs) are the statistical models of the results of multiple sequence comparisons of a given enzyme. This model provides position-specific information for each amino acid in the results of a multiple sequence comparison, as well as information on the probability of each amino acid residue at each site to have a total score in common for a given amino acid, specific motif of each enzyme. The Pfam database (*Mistry et al., 2021*) is a large collection of protein families, each represented by multiple sequence alignments and HMMs.

In this research, we present our results on the sequence and structure analyses of 31 putative LPMOs mined from the metagenomic DNA data of humus samples harvested around white-rot fungi from the national primary forest Cuc Phuong, Vietnam using HMM profiles. Additionally, the expression, purification, and enzymatic activity assay results of two LPMOs and a discussion on the potential role of studied LPMOs in the fungi-bacterial relationship are also provided.

## MATERIALS AND METHODS

### Metagenomic DNA data of humus samples

Metagenomic DNA data of bacteria in the humus samples collected from the Cuc Phuong tropical forest were created as described in *Le et al. (2022)*. The data contain about 52 Gb clean bases that yielded 2,611,883 contigs with a total length of 2,346 Mbs. Based on the assembly data, 3,884,879 bacterial protein-coding genes equivalent to 2,074 Mbs were identified using MetaGeneMark software. These identified genes were used for mining LPMOs using the HMMER.

### Using HMM profiles for the LPMO family to annotate metagenomic genes

The LPMO family was documented in the database Pfam under the code PF03067 with the HMM profile name LPMO_10. Utilizing the HMM profile, a scan for homologous sequences against the collection of novel genes in the metagenomic data of the humus

sample was executed by HMMER 3.0 hmmsearch (*Finn, Clements & Eddy, 2011*; accessed in September 2019). The result was filtered with the criteria of a domain-based score value no less than 30, profile coverage greater than 0.75, and bias/score ratio less than 0.1.

## Bioinformatics analysis

Multiple protein sequence alignment of 31 putative LPMOs was done using MUSCLE (https://www.ebi.ac.uk/Tools/msa/muscle/) to search the conserved motif of enzyme active site. InterPro (https://www.ebi.ac.uk/interpro/search/sequence/) was used to identify functional domains of proteins (accessed in April 2020).

A phylogenetic tree was constructed using MEGA-X (*Kumar et al., 2018*) with parameters of neighbor-joining method and the bootstrap value 1,000. Only catalytic domains of 31 proteins annotated as LPMOs by HMMER from metagenomic DNA data and 39 reference LPMOs, which included 14 bacterial LPMOs (AA10), 16 fungal LPMOs (AA9), one fungal LPMO (AA11), three fungal LPMO (AA13) (*Vu & Ngo, 2018*), two fungal LPMO (AA14), two fungal LPMO (AA15) (*Sabbadin et al., 2018*), and one fungal LPMO (AA16) (*Filiatrault-Chastel et al., 2019*) were used in phylogenetic tree analysis. The purpose of using these LPMOs as references was that these proteins cover both bacterial and fungal LPMOs, as well as active-cellulose LPMO and active-chitin LPMO. The list of the 39 reference LPMOs are shown in Table S2.

## Three-dimensional structure prediction of LPMOs

The three-dimensional structure of all putative LPMOs in this study was predicted using AlphaFold2 version v2.3.1+1 on Latch (https://console.latch.bio/) with the same parameters (number of models-set single and database size-set full at https://www.nature.com/articles/s41586-021-03819-2#Sec10).

Structure superimposed using the PyMOL Molecular Graphics System, Version 2.0 (Schrödinger, LLC; PyMOL): the LPMO domain of each predicted structure was superimposed with relevant reference structure using PyMOL. A particular sequence of LPMO domain was blasted against the PDB database. The most homologous structure was used as the reference structure for superimposing.

## Protein expression in *E. coli* with vector pE-SUMOpro3

Codon-optimized genes encoding the full length of mature proteins GL0183513 and GL0247266 (denoted as gene *gl* 01 and gene *gl* 02) inserted into the pET22b(+) vector were purchased from Genscript (Piscataway, NJ, USA) and were used as templates to amplify the target genes by PCR with pairs of specific primers (forward: 5′-TTGGTCTCTAGGTCATGGTTATATCCAGGATC-3′ and reverse: 5′-GCTCTAGATTAATGTTTATCCCAAGCCATC-3′ for gene *gl* 01; forward: 5′-TTGGTCTCTAGGTCATGGTGCTATGGAAAATC-3′ and reverse: 5′- CGCTCTAGATCACTGTTGTTTCCACAATTCAG-3′ for gene *gl* 02) to clone into pE-SUMOpro3 (acronym: peSUMO3) expression vector. It is noteworthy that the primers used in our work were designed to get fused proteins between SUMO and target proteins, as presented in Fig. 1. In this fusion form, the N-ends of the target proteins were fused to the C-end of the SUMO molecules. After cleavage with SUMO protease, the target proteins with His (H1) at the

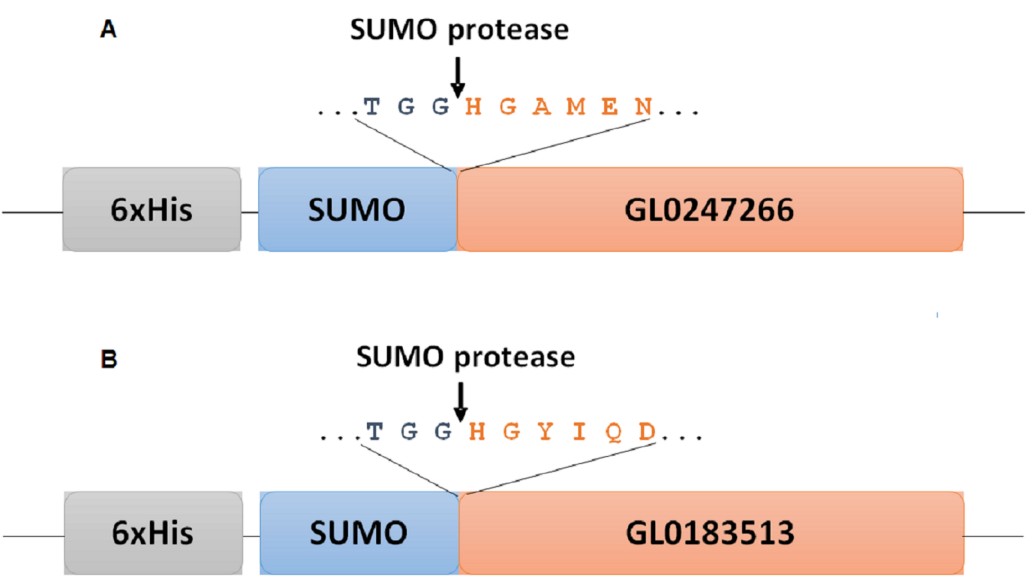

**Figure 1** **Diagram of fusion site between SUMO part and target protein in expression vectors for proteins GL0247266 (A) and GL0183513 (B) in *E. coli*.** Boxes symbolize components of fused proteins SUMO-GL0247266 and SUMO-GL0183513 with His-Tag at the N-termini. Arrows indicate the position of specific cleavage site of SUMO protease to release the mature sequence of protein GL0247266 and GL0183513 with His1 at the N-termini.

N-end of protein molecules were released. His-tag was also created before the SUMO to facilitate affinity chromatography purification with a nickel column.

The PCR products of gene *gl*01, *gl*02 and peSUMO3 vector were treated with *Bsa* I and *Xba* I, then they were cleaned separately using a KIT (MEGAquick-spinTM Plus Total Fragment DNA Purification Kit, iNtRON Biotechnology, Seongnam, Kyonggi-do, Korea) to remove short oligo fragments produced by the enzymatic cleavage reaction. After cleaning, the resulting products of purified PCR gene *gl* 01 and *gl* 02 were separately ligated into the peSUMO3 expression vector. The ligation mixtures of *gl* 01-peSUMO3 and *gl* 02-peSUMO3 were transformed into competent *E. coli* Rosetta (DE3) cells (Novagen, Merck KgAA, Darmstadt, Germany) for expression.

The transformed cells were cultured in sterilized LB medium containing 100 μg/ml ampicillin, incubated at 37 °C, and shaken at 200 rpm. When $OD_{600}$ reached a value of approximately 0.6−0.8, the expression of the recombinant fusion peSUMO-target proteins (GL02471266 or GL0183513) was induced by adding 0.2 mM IPTG for an additional 5 h at 25 °C and shaken at 180 rpm. The cells were harvested by centrifugation at 5,000 g for 10 min at 4 °C and used for protein extraction and purification by His-tag affinity column chromatography.

## Recombinant protein purification by His-tag affinity column chromatography

The cell pellets were suspended in PBS buffer 1X (137 mM NaCl, 2.7 mM KCl, 10 mM Na2HPO4, and 1.8 mM KH2PO4, pH 7) and lysed using ultrasonic sonication. The
supernatant fraction containing recombinant fusion SUMO-target protein was collected by centrifugation at 13,000 g for 15 min. The resultant supernatant was loaded onto a nickel affinity column chromatography (HiTrap) equilibrated with buffer PBS, pH 7. Non-specifically bound proteins were removed by washing with 5 column volumes of washing buffer PBS containing 20 mM imidazole. After washing, the fusion protein was eluted with elution buffer PBS containing 200 mM imidazole in the case of SUMO-GL0183513 and 300 mM imidazole for SUMO-GL0247266.

The fusion proteins SUMO-GL0183513 (S-GL0183513) and SUMO-GL0247266 (S-GL0247266) were treated with 0.2 U SUMO protease per 100 $\mu$g fusion protein at 37 °C for 3 h in the presence of 2 mM DTT to separate SUMO off the target proteins. After treating, the mixture reaction was dialyzed overnight again at 4 °C in buffer PBS 1X, pH 7.0, to eliminate imidazole and DTT. Since both SUMO and SUMO protease contain 6xHis tags, and GL0183513 and GL0247266 do not, the cleaved SUMO fusion samples could be reapplied to the nickel column to obtain the purified target proteins. Protein GL0247266 and GL0183513 were eluted at 90 mM and 100 mM imidazole by nickel affinity column chromatography, while SUMO and SUMO protease remained on the column and could be collected later by elution with 500 mM imidazole. The obtained proteins GL0183153 and GL0247266 were dialyzed against ammonium acetate buffer 50 mM, pH 6, overnight at 4 °C. After dialysis, concentration of the target proteins in solution was adjusted to 5 $\mu$M, then CuSO4 was added to final concentration 10 $\mu$M (maximum to 15 $\mu$M). The mixture was incubated for 2 h at 25 °C, then dialyzed against 50 mM ammonium acetate buffer, pH 6 overnight at 4 °C with 3 buffer changes. The purified GL0183513 and GL0247266 were checked using SDS-PAGE and the samples were stored at −20 °C for activity assay.

To determine whether GL0247266 and GL0183513 proteins still retain histidine at the N-terminus after cleaving off from SUMO, we transferred both proteins to a PVDF membrane after subjecting them to polyacrylamide gel electrophoresis. The membranes containing purified protein were then deposited to BioFarmaSpec (USA) to determine the N-terminal amino acid sequence using the Edman degradation method on the ABI Procise 494HT system (Thermo Fisher, Waltham, MA, USA).

The protein band was identified by MS/MS analysis using a nanoAcquity system (Walters, Milford, MA, USA) connected to a Orbitrap Elite hybrid mass spectrometer (Thermo Electron, Bremen, Germany) equipped with a PicoView nanospray interface (New Objective, Woburn, MA, USA). Mass spectrometry analysis database matching was carried out using PEAKS 6 software (Bioinformatics Solutions, Ontario, Canada). To identify proteins, the proteome data was taken from the UniProt Knowledgebase (UniProtKB), which was accessed in June 2023.

### Qualitative activity assay on $\beta$-chitin

Reconstitution of copper: Copper reconstitution of GL0247266 and GL0183513 was performed to coordinate Cu(II) ions with its active site as described in *Vu et al. (2019)*. The purified GL0247266 and GL0183513 solutions were treated with 10 mM EDTA to remove metal ions, followed by the removal of residual EDTA by buffer exchange using a HiTrap G-25 column. A two-fold excess of CuSO4 was slowly added to the desalted protein

solution and incubated at room temperature for 1–2 h. The protein solution was purified using a HiPrep 26/10 column and dialyzed against ammonium acetate buffer, pH 6 to completely remove free Cu(II) ions from purified GL0247266 and GL0183513. The final products, GL0247266 and GL0183513 reconstituted with Cu(II), were used in the assay enzyme activity.

Qualitative activity assay on $\beta$-chitin of proteins GL0183513 and GL0247266 was carried out with 5 µM GL0183513 or GL0247266, 5 mg/ml $\beta$-chitin, 2 mM ascorbic acid (Asc), and 10 µM CuSO$_4$ in 50 mM ammonium acetate buffer, pH 6.0, and the total volume of the reaction was 500 µl. The reaction was carried out overnight at 42 °C shaken at 800 rpm. The reaction was stopped by adding NaOH to 0.1 M. The supernatant was collected by centrifugation for 15 min at 13,000 g and diluted three to five times (depending on experiment requirements) for analysis (*Beeson et al., 2015*; *Fowler et al., 2019*). Hydrolyzed products of the reaction were monitored using HPAEC ICS-3000 (Dionex Co, Sunnyvale, CA, USA) with diacetyl-chitobiose, triacetyl-chitotriose, tetraacetyl-chitotetraose, pentaacetyl-chitopentaose, and hexaacetyl-chitohexaose (Achi$_{2-6}$) as standards.

The synergistic effect between proteins GL0183513 or GL0247266 and chitinase on $\beta$-chitin hydrolysis was detected using the following steps: reaction containing 5 µM of GL0183513 or GL0247266, 5 mg/ml $\beta$-chitin, 2 mM ascorbic acid, 10 µM CuSO4, and 7.6 U chitinase (Megazyme, Bray, Wicklow, Ireland) in 50 mM ammonium acetate buffer, pH 6.0, in total volume of 2 ml, was carried out for 4 h at 42 °C, shaken 800 rpm. The remaining steps were conducted as described above for qualitative activity assay.

## RESULTS

### Multisequence alignment of 31 putative LPMOs
Of the nearly 3.9 million ORFs of the metagenomic DNA data of the humus samples around white-rot fungi, 69 genes annotated by HMMER encoded for putative LPMOs, 33 of which were complete (*Le et al., 2022*). However, after the first round of sequence alignment, we found that only 31 proteins were complete (Table S1). The amino acid alignment analysis of the 31 putative LPMOs showed that the number of conserved amino acids among these proteins was very low, with only nine amino acid residues (H46, G47, P52, R55, H201, G254, W262, F272 and D278, according to the numeration of protein GL1034380), two histidines (H46 and H201), and one phenylalanine (F272) (Fig. 2). Alanine at position 137 (A137) of GL0247266 was also highly conserved among 29 proteins. The full amino acid sequence alignment of the 31 putative LPMOs is presented in Article S1 and Data S1.

### Phylogenetic analysis of 31 putative LPMOs
The results of the phylogenetic analysis of the 31 putative LPMOs in this study with 14 bacterial LPMOs (AA10) and 25 fungal LPMOs (AA9, AA11, AA13, AA14, AA15, and AA16) (Fig. 3) showed that fungal LPMOs were clearly separated into out-groups off the bacterial LPMOs. The 31 putative LPMOs were classified into three main groups, and Group 3 was further subdivided into six subgroups. Group 1 was comprised of five proteins and four reference LPMOs with specific cellulose activity (*Vu & Ngo, 2018*). Group 2 was

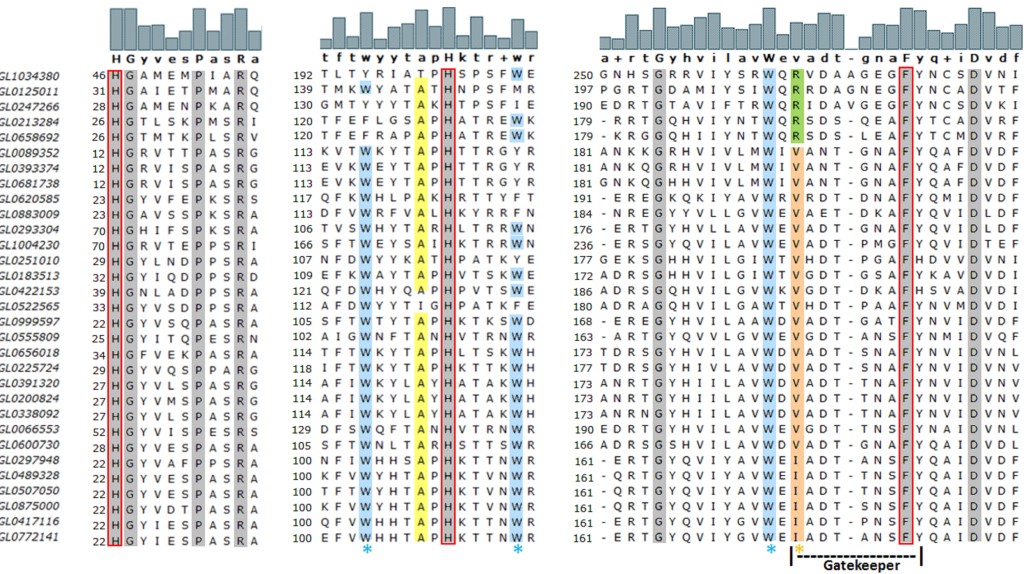

**Figure 2** **Amino acid sequence alignment of regions containing three conserved amino acids in the H1-Hx-Fy motif of 31 putative LPMOs.** Gray highlighting positions indicate conserved amino acid residues in 31 putative LPMOs. The motif H1-Hx-Fy in active site of bacterial LPMOs is highlighted in red rectangular. The orange asterisk indicates the position of amino acids specific for chitin-active bacterial LPMOs (orange highlighting isoleucine or valine) or cellulose-active bacterial LPMOs (green highlighting arginine). The blue asterisk indicates the position of the tryptophan residues that form the specific tryptophan triad in the chitin-active bacterial LPMO. The yellow highlighting alanines are specific for chitin-active bacterial LPMO. The Gatekeeper sequence "vadtgnaFy" is indicated by dot line. The numbers at the left of the columns represent the first residue position in the sequence of proteins. The logos above the protein sequences indicate the frequency of amino acid residues in the compared sequences. Amino acids bolded below the logos are consensus amino acids as determined by MEGA-X.

a distinct group that was comprised of three proteins with no reference LPMOs merged into this group. Group 3 was subdivided into six subgroups. Subgroup 3.1 contained four proteins, Subgroup 3.2 contained three proteins, Subgroup 3.3 consisted of six proteins, Subgroup 3.4 consisted of three proteins, Subgroup 3.5 contained only one protein, and Subgroup 3.6 consisted of six proteins. Three reference proteins CjLPMO10A5fjq|AA10, ScLPMO10B_4oy6|AA10, and TfE7_4gbo|AA10 did not merge with any groups of putative LPMOs, while seven remaining reference LPMOs merged with Group 3, and all these were chitin-active LPMOs (*Vu & Ngo, 2018*).

The amino acid alignment of the catalytic domains of the 31 proteins in eight groups showed that these groups had different degrees of amino acid conservation, *e.g.*, the degree of identity of Group 1 was 12%, Group 2 was 83.3%, Subgroup 3.1 was 37.5%, Subgroup 3.2 was 37.3%, Subgroup 3.3 was 37.5%, Subgroup 3.4 was 41.6%, and Subgroup 3.6 was 96.9%. With the exception of Subgroup 3.6 proteins which all had a high degree of conservation over the entire length of the protein molecules (58.7%), conserved amino acids were only observed in the catalytic domains across the remaining groups (Data S2).

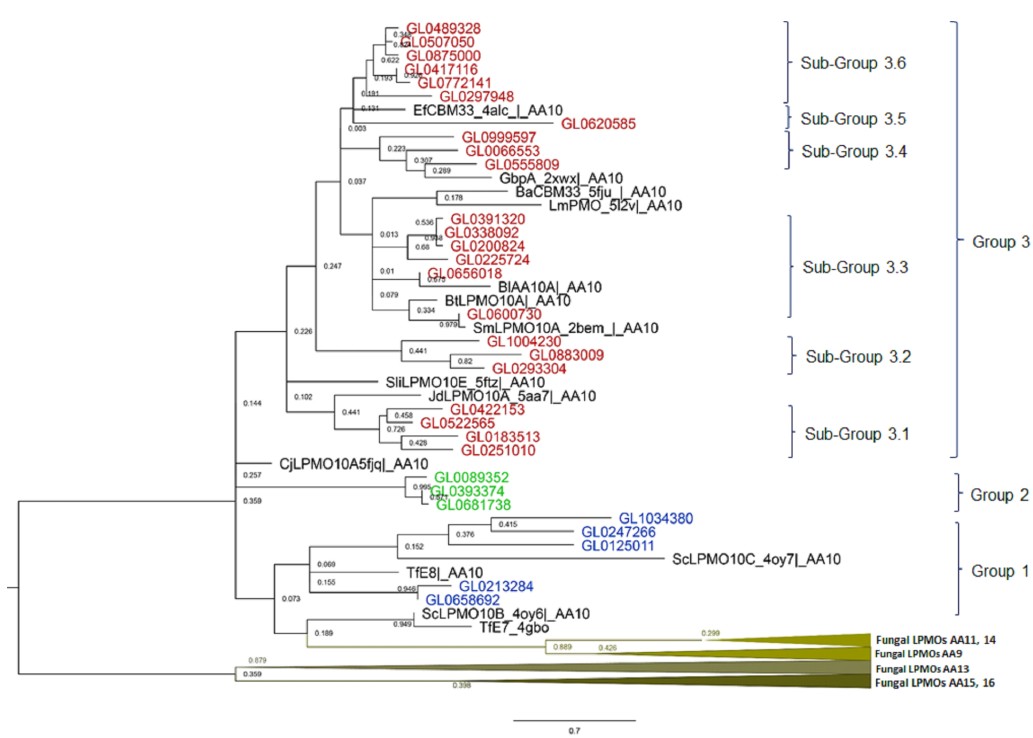

**Figure 3** **Phylogenetic tree of catalytic domains of 31 putative LPMOs, 14 bacterial LPMOs, and 25 fungal LPMOs by neighborhood-joining method MEGA-X.** The bootstrap value was set to 1,000. Blue: Group 1; Green: Group 2; Brown: Group 3; Black: Bacterial reference LPMOs. Fourteen bacterial LPMO (AA10) and 29 fungal LPMOs (16 AA9, one AA11, three AA13, two AA14, two AA15, and one AA16) were used as reference LPMOs in phylogenetic analysis.

## Prediction of functional domains of 31 putative LPMOs

The functional domain of 31 putative LPMOs was predicted using the InterPro tool. The results presented in Fig. 4A show that the 31 proteins in this study had a very complex structure. Group 2 proteins consisted of a single catalytic domain, while proteins of other groups consisted of two to five functional domains. Besides catalytic (LPMO) domain, other functional domains were discovered in these proteins, such as GbpA_2, Fn3, Ig-like, CBMs (CBM5, CBM73 and CBM 5/12), and Secret_tail_C. The lengths of functional domains among protein groups also varied, for example, the catalytic domain ranged from 151 to 197 amino acids, the GbpA domain 90–98 amino acids, the Ig-like domain 88–195 amino acids, FN3 domain 90–186 amino acids, and the CBM_sf_5/12 domain 39–92 amino acids. The CBM5 domain had a constant length (40 amino acids). Three proteins, GL0391320, GL033892, and GL0200824 (Subgroup 3.3), had a special structure of two consecutive CBM5 domains. In addition, proteins GL0391320 and GL0200824 contained the C-terminal Secret_tail_C.

The proteins GL0247266 (Group 1), GL0183513, GL0251010, GL0522565, GL0422153 (Subgroup 3.1), GL0338092 (Subgroup 3.3), GL0555809, GL0066553 (Subgroup 3.4), GL0620585 (Subgroup 3.5), and all six proteins of Subgroup 3.6 contained a region that

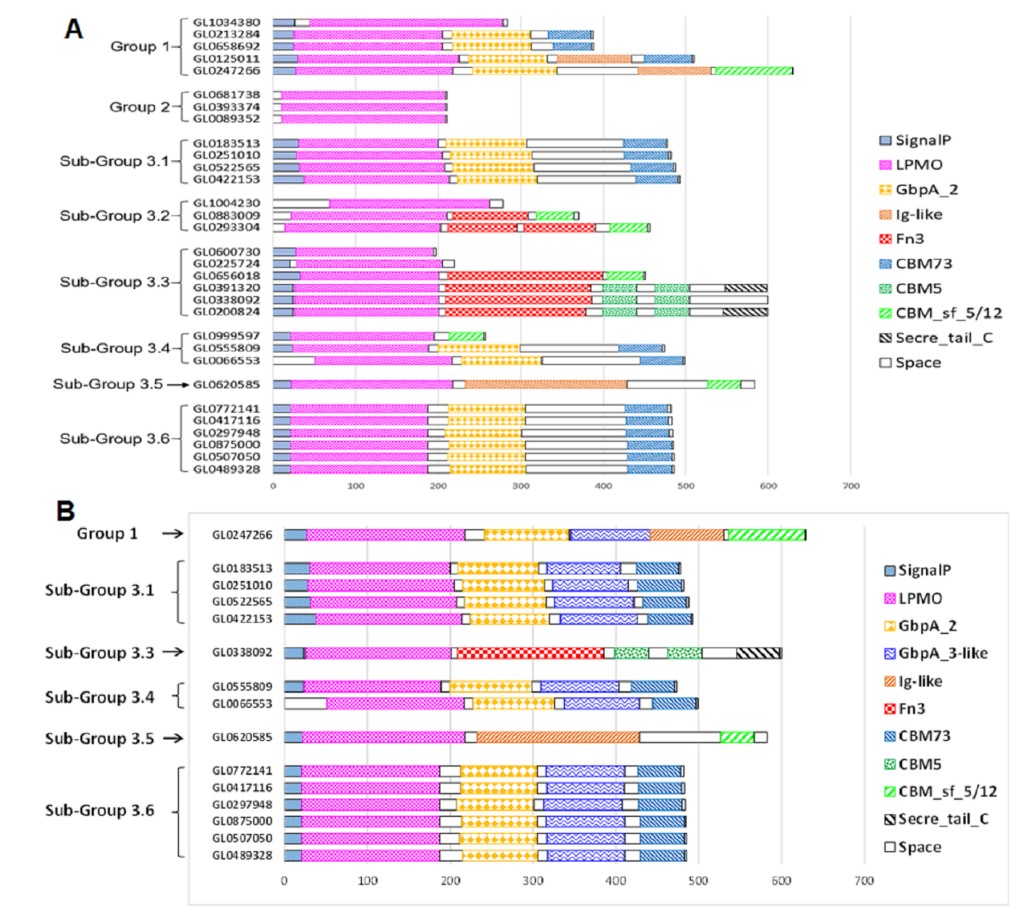

**Figure 4** **Prediction of functional domains of 31 putative LPMOs.** (A) Prediction of functional domains of 31 putative LPMOs using InterPro. (B) Prediction of GbpA_3-like domains in 13 putative LPMOs using Alphafold2. Functional domains are represented by boxes with different patterns and colors.

was annotated by InterPro as a non-functional region (denoted as "Space" region in Fig. 4A).

## Structure characteristics of 31 putative LPMOs

Structural analysis of the 31 proteins revealed that they differed in the amount of cysteine in the catalytic domain of the enzyme (Table 1). Some proteins (GL0658692 and GL021328 of Group 1) contained six cysteines in the catalytic domain capable of forming three disulfide bridges. Other proteins (GL0247266 and GL0125011 of Group 1, all proteins of Subgroup 3.1, protein GL06000730 of Subgroup 3.3, all proteins of Subgroup 3.4, and all proteins of Subgroup 3.6) contained four cysteines capable of forming two disulfide bridges. The others contained two cysteines (GL0656018 of Subgroup 3.3 and GL0620585 of Subgroup 3.5) or three cysteines (GL1034380 of Group 1) capable of forming a disulfide bridge. Meanwhile, some proteins did not contain any cysteine residues (GL0293304 of Subgroup 3.1; protein GL0293304 of Subgroup 3.2; and proteins GL0225724, GL0391320, GL0200824, GL0338092, and GL0600730 of Subgroup 3.3) or contained only one residue

**Table 1** Structure characteristics of 31 putative LPMOs.

| Groups | Proteins | Length of full/mature proteins (aa) | Positions of H1, Hx, Fy in catalytic domain | Number of cysteines in full-length protein/catalytic domain |
|---|---|---|---|---|
| Group 1 | GL0247266 | 631/604 | $H_1$-$H_{112}$-$F_{185}$ | 4/4 |
| | GL1034380 | 284/240 | $H_1$-$H_{156}$-$F_{227}$ | 4/3 |
| | GL0125011 | 510/481 | $H_1$-$H_{118}$-$F_{189}$ | 6/4 |
| | GL0658692 | 389/365 | $H_1$-$H_{104}$-$F_{174}$ | 8/6 |
| | GL0213284 | 388/364 | $H_1$-$H_{104}$-$F_{174}$ | 8/6 |
| Group 2 | GL0393374 | 211/201 | $H_1$-$H_{111}$-$F_{191}$ | 1/1 |
| | GL0681738 | 211/201 | $H_1$-$H_{111}$-$F_{191}$ | 1/1 |
| | GL0089352 | 211/201 | $H_1$-$H_{111}$-$F_{191}$ | 1/1 |
| Sub-Group 3.1 | GL0183513 | 478/448 | $H_1$-$H_{87}$-$F_{162}$ | 6/4 |
| | GL0251010 | 483/456 | $H_1$-$H_{88}$-$F_{170}$ | 6/4 |
| | GL0522565 | 488/457 | $H_1$-$H_{126}$-$F_{169}$ | 6/4 |
| | GL0422153 | 493/456 | $H_1$-$H_{92}$-$F_{169}$ | 6/4 |
| Sub-Group 3.2 | GL1004230 | 279/211 | $H_1$-$H_{106}$-$F_{187}$ | 4/4 |
| | GL0883009 | 370/349 | $H_1$-$H_{99}$-$F_{182}$ | 2/1 |
| | GL0293304 | 457/443 | $H_1$-$H_{100}$-$F_{181}$ | 1/0 |
| Sub-Group 3.3 | GL0656018 | 451/419 | $H_1$-$H_{90}$-$F_{161}$ | 2/2 |
| | GL0225724 | 220/193 | $H_1$-$H_{99}$-$F_{170}$ | 0 |
| | GL0391320 | 599/574 | $H_1$-$H_{97}$-$F_{168}$ | 8/0 |
| | GL0200824 | 600/575 | $H_1$-$H_{97}$-$F_{168}$ | 7/0 |
| | GL0338092 | 600/575 | $H_1$-$H_{97}$-$F_{168}$ | 8/0 |
| | GL0600730 | 197/171 | $H_1$-$H_{87}$-$F_{160}$ | 4/4 |
| Sub-Group 3.4 | GL0066553 | 499/449 | $H_1$-$H_{87}$-$F_{160}$ | 6/4 |
| | GL0555809 | 474/451 | $H_1$-$H_{87}$-$F_{159}$ | 6/4 |
| | GL0999597 | 258/238 | $H_1$-$H_{93}$-$F_{167}$ | 4/4 |
| Sub-Group 3.5 | GL0620585 | 583/562 | $H_1$-$H_{104}$-$F_{189}$ | 4/2 |
| Sub-Group 3.6 | GL0297948 | 484/464 | $H_1$-$H_{88}$-$F_{160}$ | 6/4 |
| | GL0875000 | 485/465 | $H_1$-$H_{88}$-$F_{160}$ | 6/4 |
| | GL0489328 | 486/466 | $H_1$-$H_{88}$-$F_{160}$ | 6/4 |
| | GL0507050 | 486/466 | $H_1$-$H_{88}$-$F_{160}$ | 6/4 |
| | GL0417116 | 483/463 | $H_1$-$H_{88}$-$F_{160}$ | 6/4 |
| | GL0772141 | 482/462 | $H_1$-$H_{88}$-$F_{160}$ | 6/4 |

(all proteins of Group 2 and protein GL0883009 of Subgroup 3.2) in the catalytic domain. These proteins were incapable of forming disulfide bridges in the catalytic domain.

## Prediction of spatial arrangement of conserved histidines and phenylalanine in active site of 31 putative LPMOs

Based on the clustering of 31 proteins in the phylogenetic tree, we predicted the 3-D structure of catalytic domain of eight group proteins using Alphafold2 and spatially compared it with each other using PyMOL. The results showed that the protein catalytic
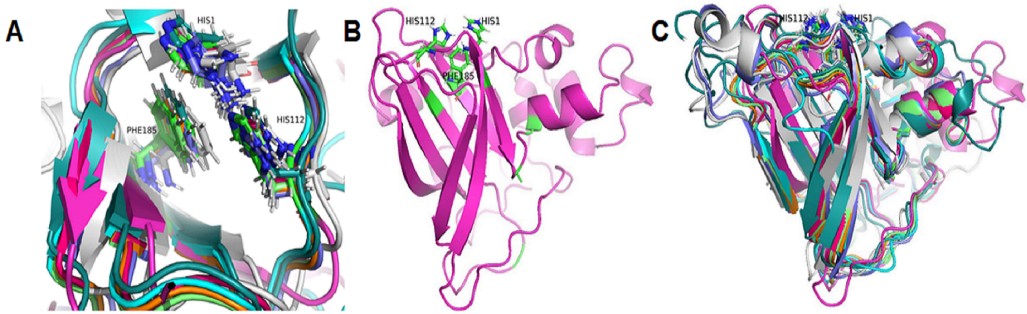

**Figure 5  Three-dimensional structure prediction of catalytic domains and spatial arrangement of conserved histidine brace and phenylalanine in active site of putative LPMOs by Alphafold2.** (A) Spatial arrangement of two histidines and phenylalanine in active sites of representatives of eight phylogenetic groups. (B) 3-D structure of GL0247266. (C) Spatial structure comparison of catalytic domains of representative of eight phylogenetic groups. Representative proteins: GL0247266 (Group 1) - magenta, GL0083952 (Group 2) - gray, GL0183513 (Subgroup 3.1) - cyan, GL0293304 (Subgroup 3.2) - slate, GL0600730 (Subgroup 3.3) - orange, GL0066553 (Subgroup 3.4) - lime green, GL0620585 (Subgroup 3.5) - deep teal, GL0297948 (Subgroup 3.6) - warm pink.

domains in each group had very similar 3-D structures (Data S3). Therefore, to simplify the visualization of the comparison results, only eight proteins representing eight phylogenetic groups (GL0247266 of Group 1, GL0083952 of Group 2, GL0183513 of Subgroup 3.1, GL0293304 of Subgroup 3.2, GL0600730 of Subgroup 3.3, GL0066553 of Subgroup 3.4, GL0620585 of Subgroup 3.5, and GL0297948 of Subgroup 3.6) were compared according to the arrangement of conserved histidines, phenylalanine residues in the active site, and 3-D structure of the catalytic domains. The results showed that the conserved histidines and phenylalanine residues in the active site of these eight proteins were very similar in spatial arrangement (Fig. 5A, Article S2, Data S4) and it is similar to the arrangement of these residues of previously described LPMOs.

Additionally, the two conserved histidines and phenylalanine of the proteins in each group were also compared with those of the reference LPMOs. The comparison showed that the arrangements of the two conserved histidines and phenylalanine in the active site of the LPMOs in each group with the known substrate-specific reference LPMOs were very similar. Specifically, the spatial arrangement of the two histidines and phenylalanine in the active site of the Group 1 proteins resembled the spatial arrangement of the H43, H150, and F219 residues of the active site of the cellulose-active LPMO (4OY6) from *S. coelicolor* A3 (*The Protein Data Bank, 2023a*). Similarly, the spatial arrangement of the two histidines and phenylalanine in the active site of the Group 2 and Group 3 proteins resembled the spatial arrangement of those in the active site of the 3UAM of *Burkholderia pseudomallei* 1710b (*The Protein Data Bank, 2023b*), 5AA7 from *Jonesia denitrificans* (*The Protein Data Bank, 2023c*), 5LW4 from *B. licheniformis* (*The Protein Data Bank, 2023d*), and 2BEM from *S. marcescens* (*Vaaje-Kolstad et al., 2005a*; *Vaaje-Kolstad et al., 2005b*) (Data S2). It is important to note that the proteins 3UAM, 5AA7, 5LW4, and 2BEM were chitin-active LPMOs. The results suggest that the 31 proteins were potential LPMOs and

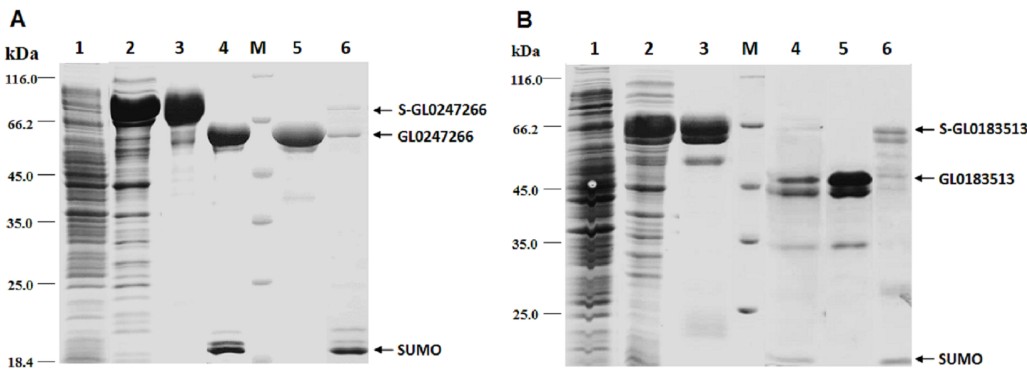

**Figure 6** **Results of protein expression in *E. coli* by vector pET-SUMO3 and protein purification using His-tag affinity chromatography.** 1: Non-induced total protein fraction; 2: Induced total soluble protein fraction by IPTG; 3: Fused SUMO-GL0247266 (A) and SUMO-GL0183513 (B); 4: Cleavage products of SUMO-GL0247266 (A) and SUMO-GL0183513 (B) treated with SUMO protease; 5: Protein GL0247266 eluted at 90 mM imidazole (A) and GL0183513 eluted at 100 mM imidazole (B). 6: SUMO fraction eluted at 500 mM imidazole; M: Protein molecular mass standards (Fermentas).

may have different substrate-specific activities. Specifically, Group 1 proteins may have cellulose-specific activity and Group 2 and 3 proteins may have chitin-specific activity.

After comparing the 3-D structures of representative proteins of eight phylogenetic groups, we found that these proteins all had seven $\beta$-strands in the catalytic regions (Fig. 5B) and their 3-D structures were very similar in spatial shape (Fig. 5C, Data S4). This means that although the 31 proteins had very diverse amino acid sequences, they all had very similar 3-dimensional configuration.

## Expression and purification of proteins GL0247266 and GL0183513 in *E. coli*

The genes coding proteins GL0247266 and GL0183513 were expressed in *E. coli* using vector peSUMO3 and purification as described in the Methods. Both proteins GL0247266 and GL0183513 were highly expressed in *E. coli* in fusion with SUMO, specifically, fusion proteins S-GL0247266 (lane A2, Fig. 6) and S-GL0183513 (lane B2, Fig. 6). The fused proteins S-GL0247266 and S-GL0183513 were purified using His-tag affinity chromatography (lane A3 and B3, Fig. 6). After treating with SUMO protease to cleave off the SUMO part, proteins GL0247266 and GL0183513 were re-purified using His-tag affinity chromatography (lanes A4 and B4, respectively; Fig. 6). The protein GL0247266 (lane A5, Fig. 6) was nearly completely purified, however, the protein GL0183513 (lane B5, Fig. 6) could not be purified completely by His-tag affinity chromatography. There was a second smaller protein band that accompanied the GL0183513 protein. MS/MS analysis of this smaller protein band indicated that this protein was highly likely to be a chaperone. The results showed that matching was highest to Chaperone HtpG from *Escherichia coli* (Table S2). Therefore, although the GL0183513 protein was not completely purified, it can still be used to determine qualitative activity of LPMO.

To determine whether GL0247266 and GL0183513 proteins still retained histidine at the N-terminus after cleaving off SUMO using SUMO protease, both proteins were deposited

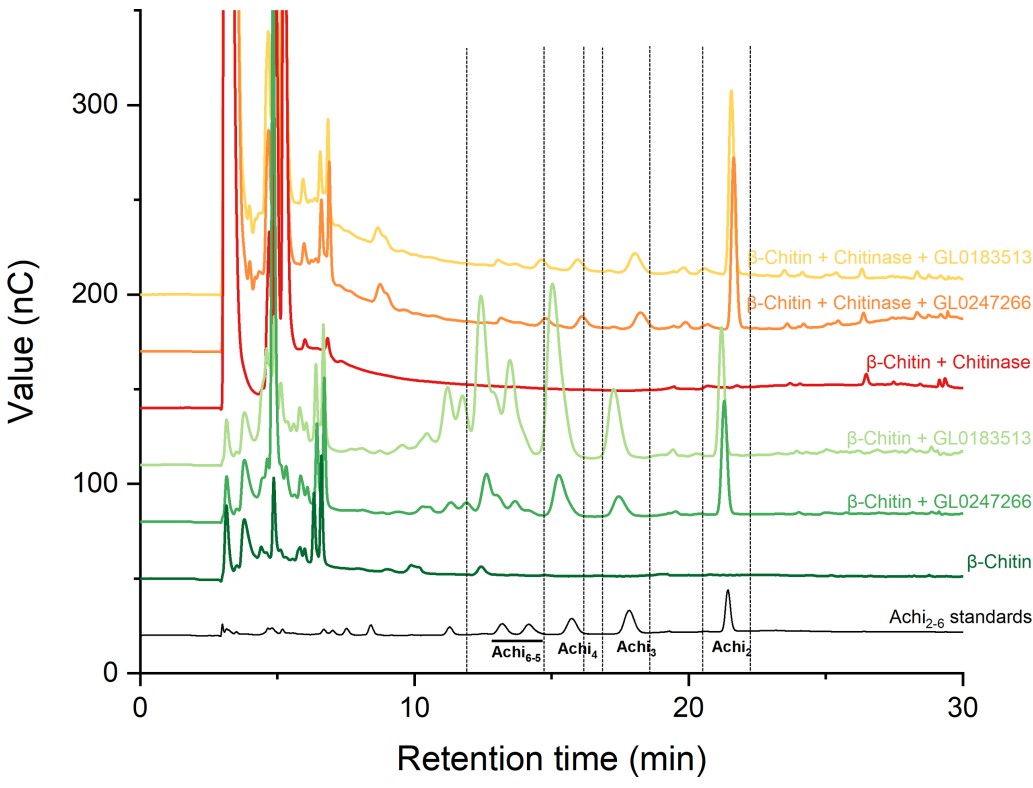

**Figure 7  HPAEC-PAD chromatograms for LPMO activity assay with $\beta$-chitin of GL0183513 or GL0247266 alone (green) and in combination with chitinase (red).** Key products derived from LPMO activity (Achi$_2$, Achi$_3$, Achi$_4$ and Achi$_{5-6}$) are indicated within dashed rectangles. Reactions contained 5 $\mu$M GL0183513 or GL0247266, 5 mg/ml $\beta$-chitin, 2 mM AscA, and 10 $\mu$M CuSO$_4$ in 50 mM ammonium acetate buffer, pH 6.0, and were incubated at 42 °C and 800 rpm for overnight. Dark green and dark red chromatograms show $\beta$-chitin and $\beta$-chitin treated with chitinase, respectively, incubated with AscA and without GL0183513 or GL0247266. Standards consisting of oligomers from Achi$_{2-6}$ are shown in black.

to BioFarmaSpec (USA) to determine the N-terminal five amino acid sequence by Edman degradation method. Results of N-terminal amino acids sequencing showed that the first five amino acids of N-terminal of GL0247266 were HGAME, while the five first amino acids of N-terminal of GL0183513 were HGYIQ, as expected (Article S3).

## Qualitative activity assay on $\beta$-chitin

Two proteins GL0247266 and GL0183513 were tested with cellulose and chitin substrates to analyze enzymatic activity. Results showed that both proteins have no LPMO activity with cellulose (Fig. S1); however, they have LPMO activity with $\beta$-chitin. The results presented in Fig. 7 show that GL0247266 (forest green) and GL0183513 (light green) were capable of oxidizing $\beta$-chitin to produce diacetyl-chitobiose, triacetyl-chitotriose, tetraacetyl-chitotetraose, pentaacetyl-chitopentaose, and hexaacetyl-chitohexaose as products.

Next, these proteins were also mixed with chitinase during $\beta$-chitin hydrolysis to check if they showed synergistic action. The results presented in Fig. 7 showed that the combination of protein GL0247266 (orange) or GL0183513 (yellow) with chitinase made

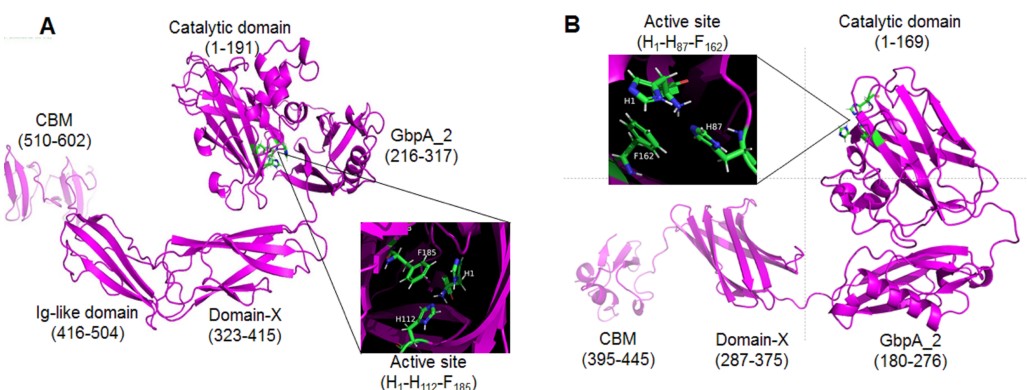

**Figure 8  Three-dimensional structures of mature GL0247266 (A) and GL0183513 (B) predicted using Alphafold2.** Functional domain locations are marked. The structure of the active site is zoomed in in the boxes.

chitinase hydrolysis of $\beta$-chitin occur more thoroughly when the main product in the reaction was diacetyl-chitobiose, the end product of chitinase upon $\beta$-chitin hydrolysis. Meanwhile, chitinase alone hardly produces $\beta$-chitin hydrolysis product (red). This evidence demonstrated the synergy between LPMOs (GL0247266 and GL0183513) and chitinase in $\beta$-chitin depolymerization. These assays were the final confirmation that both GL0247266 and GL0183513 proteins are the real chitin-active LPMOs.

## Prediction of 3-dimensional structure of GL0247166 and GL0183513 by Alphafold2

The proteins GL0247266 and GL0183513 were further studied on the 3-dimensional structure using the Alphafold2 tool. The 3-dimensional structure prediction of proteins GL0247266 and GL0183513 presented in Fig. 8 shows that their functional domains were folded in specific spatial structures. Specifically, the catalytic domain (1 to 191) of the protein GL0247266 (Fig. 8A) had the specific structure of LPMO with seven beta-strands and the active site located in the loops that connected the $\beta$-strands and exposed on the surface of the protein molecule. In addition to the catalytic domain, other domains such as GbpA$_2$-domain (216–317), Ig-like domain (416-504) and CBD (510–602) also had distinct spatial structures. Similarly, the catalytic domain (amino acids 1–169), GbpA$_2$-domain (180–276), and CBD (395–445) of protein GL0183513 (Fig. 8B) also had specific spatial structures.

Interestingly, the protein sequences from amino acid 317 to 416 of protein GL0247277 and from 276 to 395 of protein GL0815513 were predicted by InterPro to be non-structural domains (space regions in Fig. 4A). However, Alphafold2 predicted that these two regions in proteins GL0247266 (92 amino acids) and GL0183513 (88 amino acids) had a compact structure with the topology of a twisted $\beta$-sandwich which contained two antiparallel $\beta$-sheets, one composed of three $\beta$-strands, and the other of four $\beta$-strands. These two regions of unknown function in the proteins GL0247266 and GL0183513 were denoted as Domain-X.
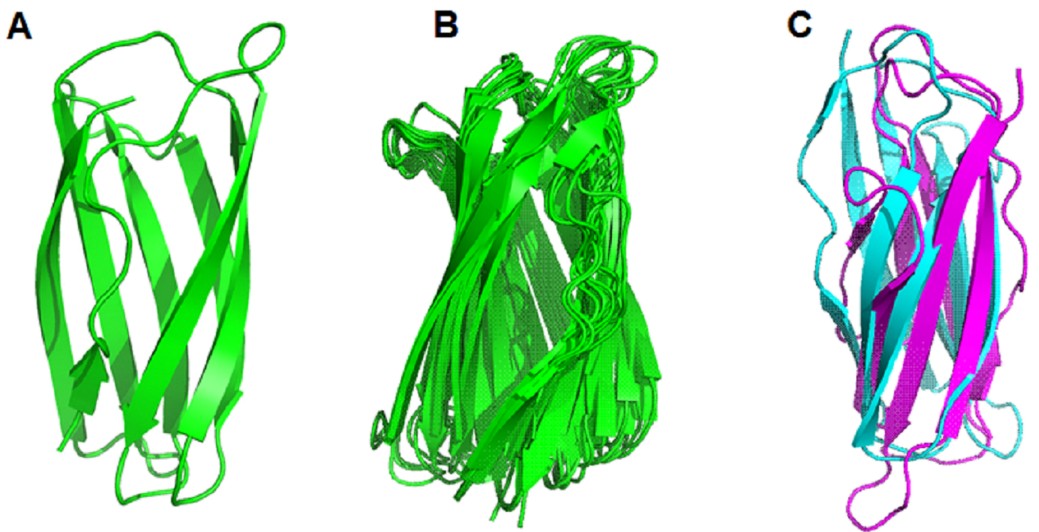

**Figure 9** **Comparison of the spatial structure of the Domain-X of 13 putative LPMOs with the GbpA_3 domain of *Vibrio cholera* GbpA.** (A) Three-dimensional structure of Domain-X of protein GL0251010. (B) Spatial structure comparison of Domain-X of 13 proteins GL0251010, GL0247266, GL0422153, GL0183513, GL0522565, GL0555809, GL0066553, GL0297948, GL077214, GL0417116, GL0875000, GL0489328, and GL0507050. (C) Spatial structure comparison of GL0183513 Domain-X (magenta) with the GbpA_3 domain of *V. cholerae* GbpA (Blue).

Surprisingly, the non-structural regions (space regions) in the other 11 proteins (GL0251010, GL0522565, GL0422153, GL0555809, GL0066553, GL0772141, GL0417116, GL0297948, GL0875000, GL0507050, and GL0489328) also showed spatial structures that were similar to the Domain-X of proteins GL0247266 and GL0183513, as predicted by Alphafold2. Comparing the spatial structure by superimposing the Domain-X of these 13 proteins showed that they completely coincide in spatial structure (Fig. 9B, Data S5). It is also interesting that all the Domain-X in the 13 proteins were always located immediately adjacent next to the GbpA_2 domain. This is similar to the structure of domain GbpA_3 of *V. cholera* GbpA, which has a similar compact β-strands fold structure of six long β-strands and two short β-strands (*Wong et al., 2012*). The structure comparison by the PyMOL showed that the overall topology of Domain-X of GL0183513 was similar to that of the domain 3 (GbpA_3 domain) of *V. cholera* GbpA protein (Fig. 9C, Data S6). Based on this similar spatial conformation, the Domain-X was named GbpA_3-like domain.

The results of 3-D prediction by Alphafold2 showed that the non-structural region of the GL0620585 did not fold in any specific spatial structure. Meanwhile the C-terminal region of the GL0338092 protein folded in the same spatial structure as the Secret_tail_C region of the two proteins GL0391320 and GL0200824 in the same group of Group 3.3 (Fig. 4B).

## DISCUSSION

### Sequence analysis of 31 putative LPMOs

The histidine brace is a conserved structure in all LPMOs regardless of origin and substrate specificity. In addition to the histidine brace, conservative phenylalanine plays a role in increasing the stronger affinity of the active site for Cu(II) in bacterial LPMOs (AA10) (*Beeson et al., 2015*; *Bissaro et al., 2018b*). Combining two conserved histidines with a phenylalanine can be used as the marker (H1-Hx-Fy motif) to recognize active sites of bacterial LPMOs. Structural comparisons (*Forsberg et al., 2014b*) showed that the chitin-active LPMOs contained a cavity adjacent to the active site while cellulose-active LPMOs did not. This cavity was well suited to accommodate the N-acetyl group present in the structure of chitin. EPR spectra indicated that the several residues close to the copper binding site were different between the cellulose-active and chitin-active LPMOs. In chitin-active CBP21, the isoleucine 180 (I180) was an important determinant for substrate specificity. The I180 was conserved or replaced by valine (V) in other chitin-active LPMOs of the AA10 family. This residue formed a cavity on the enzyme surface in the direct proximity of the active site. The cellulose-active LPMOs of the AA10 family contained an arginine at this position. Besides that, there were three conserved tryptophan residues located directly below the copper binding site in the chitin-active LPMOs (*Forsberg et al., 2014a*). It was suggested that the tryptophan triad plays a role in the electron transfer of LPMO in oxidation (*Hemsworth, Davies & Walton, 2013*).

In our study, amino acid sequence comparison results showed that 31 interested proteins contained conserved amino acids that are specific for the active LPMO site, such as two histidines and a phenylalanine. The presence of these three amino acids and the reasonable distance between them in all 31 proteins suggest that they may participate in the H1-Hx-Fy motif of the LPMO active site.

The substrate specificity of these proteins can be observed based on the presence of arginine (cellulose-specific activity) in the proteins GL0125011, GL0213284, GL0247266, GL1034380, and GL0658692, or isoleucine and valine (chitin-specific activity) in 26 remaining proteins (position marked with a red asterisk in Fig. 2). It is notable that these five proteins separated in Group 1 of the phylogenetic tree where four cellulose-active reference LPMOs (4oy6, 4oy7, TfE7, and TfE8) were also merged together. Meanwhile, 10 reference chitin-active LPMOs were interlocated amongst the proteins of Groups 2 and 3 (Fig. 3). The substrate specificity can be also recognized through the presence of the tryptophan triad, specifically, out of 31 proteins, 21 proteins contained all three tryptophan residues (marked with blue asterisks in Fig. 2). Based on the amino acid sequences and phylogenetic analysis, Group 1 proteins might be specific for cellulose-active LPMOs, while proteins of Group 2 and Group 3 are specific for chitin-active LPMOs.

Amino acid sequence alignment also showed a highly conserved alanine situated in front of the active-site second histidine (Hx) of 29 proteins. This alanine is supposedly similar to A112 in CBP21 (chitin-active LPMO) that was exposed closely to the active enzyme site (*Forsberg et al., 2014b*) and also highly conserved in other LPMOs (*Agostoni, Hangasky & Marletta, 2017*). The motif mainly consisted of polar amino acids (Y[W]E[N]PQSV[L]E)

detected in chitin-active bacterial LPMOs (*Zhou & Zhu, 2020*) and was also observed in 20 putative LPMOs (Article S1, Data S1). The Gatekeeper sequence ''vadtgnaFy'' that determines the substrate specificity of bacterial LPMOs (*Votvik et al., 2023*) was also detected in many putative LPMOs (Fig. 2).

## Structure analysis of 31 putative LPMOs

The structure of the 31 putative LPMOs was highly diverse from a single catalytic domain to multi-functional domains. This diversity was detected not only between proteins of phylogenetic groups but also within groups (Fig. 4A). Besides the catalytic domains, different functional domains were discovered in these proteins, such as GbpA_2, Fn3, Ig-like, CBMs (CBM5, CBM73, and CBM 5/12), and Secret_tail_C. The multi-modular structure was also detected in other LPMOs (*Wong et al., 2012*; *Hemsworth et al., 2014*; *Agostoni, Hangasky & Marletta, 2017*; *Courtade & Aachmann, 2019*). Functions of these domains were partially clarified. Although the catalytic domain of LPMOs was capable of binding to the substrates, CBMs can increase the binding capacity of LPMOs to various polysaccharides (*Boraston et al., 2004*; *Gilbert, Knox & Boraston, 2013*; *Mutahir et al., 2018*), and affect the precise positioning of the enzymes on the substrate surface relative to the position of C1 or C4 carbon to the catalytic center Cu (*Cuskin et al., 2012*; *Crouch et al., 2016*), thereby modulating the activity of enzymes (*Arfi et al., 2014*). For example, CBM5 was suggested to play an important role in hydrolysis by moving the enzyme in close proximity of substrates and the presence of CBM5 in LPMOs has been shown to alter the product profile while acting on crystalline $\beta$-chitin (*Manjeet et al., 2019*). CBM5, CBM12, and CBM73 strongly promote targeting and binding of LPMO to crystalline $\alpha$- and $\beta$-chitin (*Wong et al., 2012*; *Forsberg et al., 2016*; *Votvik et al., 2023*). Fn3 is thought to be able to support protein-protein binding and interactions (*Potts & Campbell, 1996*), or aid in the hydrolysis of cellulose by modifying its surface (*Kataeva et al., 2002*). Ig-like domains are involved in a variety of functions, including recognition, cell–surface receptors, muscle structure, and immune system (*Teichmann & Chothia, 2000*). GbpA_2 is required when binding to the bacterial surface (*Wong et al., 2012*). Secretion system C-terminal sorting domain (Secret_tail_C) targets proteins to type IX secretion systems and is secreted following cleavage by a C-terminal signal peptidase (*De Diego et al., 2016*).

Structural analysis of 31 proteins also showed that they were very diverse not only in protein lengths, but also in the number of cysteine residues. This divergence was observed not only between groups, but also between proteins inside several particular groups, *e.g.*, Group 1, Subgroup 3.2 and Subgroup 3.3 (Table 1). The difference in the amount of disulfide bonds reflects divergence in the thermostability of these proteins, since disulfide bonds are essential for the thermodynamic stability of LPMO (*Beeson et al., 2015*), and breaking disulfide bonds will cause irreversible unfolding of the protein leading to loss of enzyme activity and introduction of additional disulfide bonds may increase the thermostability of the enzyme (*Tanghe et al., 2017*).

The prediction of the three-dimensional structure of the catalytic domains of 31 putative LPMOs show that they are composed of seven long $\beta$-strands and five $\alpha$-helix chains creating a spatial configuration similar to that described in *Beeson et al. (2015)*,
(Fig. 5B). Comparison of the spatial configuration of these proteins shows that although their amino acid sequences are very different, especially the number of cysteine residues in their catalytic domains, all these proteins have very similar spatial configurations (Fig. 5C), even conservative amino acids in an active site such as two histidines of the histidine brace and phenylalanine also have very similar spatial arrangements (Fig. 5A). Our results provide additional evidence for the opinion that although PMO family members typically share low sequence identity, all structures identified to date show a high degree of spatial structure similarity (Book et al., 2014; Beeson et al., 2015; Vaaje-Kolstad et al., 2017). Here, the determinant of the conserved spatial structure of LPMOs may be the beta chains that are capable of creating complex and stable core structures in protein molecules (Youkharibache et al., 2018). In this case, evolution preferentially screened the function of LPMOs based on the spatial structure matching their substrates (recalcitrant polysaccharides), rather than the amino acid sequence of the protein. The spatial structure of the LPMOs is important for enzyme function, while substrate specificity (cellulose and chitin) is determined at the amino acid sequence level.

**Expression, purification, and LPMO activity of GL0247266 and GL0183513**
In this study, two genes were selected for expression: protein-coding gene GL0247266 (Group 1) and GL0183513 (Subgroup 3.1). These two genes were selected for expression because they were predicted to be highly expressed in *E. coli* by Periscope software (http://lightning.med.monash.edu/periscope/index.jsp) (results not presented here). Initially, these two genes were expressed in *E. coli* using vector pET22b(+), however most of the proteins received were fallen in inclusion bodies, so we switched to peSUMO3 vector expression. This vector was used to successfully express AA10 LPMO from *B. amyloliquefaciens* intracellularly in *E. coli* being fused to an N-terminal SUMO tag, yielding a functional protein after specific cleavage of the tag adjacent to the LPMO domain's active site H1 by SUMO protease treatment (Gregory et al., 2016).

It was surprising that the GL0247266 protein, according to amino acid sequence analysis (Fig. 2), phylogenetic tree analysis with reference LPMOs (Fig. 3), and active site structural similarity with reference LPMOs (Data S2), is predicted to be LPMO active with cellulose. However, our activity assay showed that this protein is a chitin-active LPMO. To elucidate this issue, the amino acid sequence of the protein GL0247266 was re-analyzed and it was found that near arginine 233, not far from phenylalanine 241 (Fig. 2) there is an isoleucine at position 234. These amino acids are importantly determinant for AA10 LPMO substrate specificity, similar to isoleucine 180 in CBP21, as previously mentioned. Arginine at this site is specific for cellulose activity, while isoleucine or valine is specific for the chitin specificity of AA10 LPMOs (Forsberg et al., 2014b). This means that the GL0247266 protein in terms of amino acid sequence of the catalytic domain can act on both cellulose and chitin substrates. However, since protein GL0247266 contains an additional C-terminal CBM_sf_5/12 domain that promotes binding to chitin (Ikegami et al., 2000; Votvik et al., 2023), so in this case, it is very likely that this functional domain favors the chitin activity of this protein.

It is known that LPMOs tend to lose activity under turnover conditions and their operational stability may be low due to self-inactivation (*Bissaro et al., 2017*). LPMOs that are reduced and meet $O_2$ or $H_2O_2$ while not being bound to a substrate are particularly susceptible to such autocatalytic inactivation, due to the reactivity of the reduced Cu(I) ion in the LPMO active site (*Kont et al., 2020*; *Forsberg & Courtade, 2023*). The inactivation of LPMO depends on the level of $H_2O_2$ generated *in situ* (*Stepnov et al., 2021*; *Stepnov, Eijsink & Forsberg, 2022*). In this case CBM can influence LPMO activity. On one hand, CBM would contribute to making the LPMO catalytic domain closer to the substrate, thus increasing the possibility of encountering effectively utilized $H_2O_2$. On the other hand, the oxygenase activity of LPMO is an important contributor to local $H_2O_2$ generation, which can be hindered by substrate binding (*Courtade et al., 2018*; *Filandr et al., 2020*; *Forsberg & Courtade, 2023*). It was also shown that the inactivation of LPMOs is independent of the nature of the reductant (*Kuusk, Eijsink & Väljamäe, 2023*). In our study, besides two proteins GL0183513 and GL0247266 containing CBM domains that were tested for enzymatic activity, most of the remaining proteins contained different CBM domains such as CBM5, CBM5/12 and CBM73 (Fig. 4). It is likely that these proteins will be stable in terms of oxidative activity under the influence of $O_2$ and $H_2O_2$.

Taxonomic classification based on blastall NR showed that the GL0247266 and GL0183513 proteins were derived from the Enterobacter and Pseudomonas genera, respectively (Table S3). Therefore, the protein GL0247266 is denoted as metE.1_LPMO_AA10 and the protein GL0183513 is metP.1_LPMO_AA10. Prefix "met" means metagenomic origin of proteins, letters "E" and "P" mean name of genera. Table S3 also shows that most of the proteins in this study belong to the phylum Proteobacteria, except for proteins of Subgroup 3.3 which belong to the phyla Bacteroidetes and Firmicutes.

## Possible role of chitin-active GbpA-like LPMOs in the relationship between fungi and bacteria

Our research involves the unique ecosystem of white-rot fungi and the bacteria that co-inhabit decomposing trees in primary forests. Between these two biological communities, in addition to different types of symbiotic interactions such as cooperation and mutualism, there is also antagonism and nutritional competition. One of the mechanisms by which fungi can control the growth of bacteria is to change the pH in the surrounding environment (*Johnston et al., 2018*; *Embacher et al., 2023*). Conversely, bacteria can suppress fungi by synthesizing cell wall-degrading enzymes (LPMO and chitinase). Bacterial LPMOs with chitin activity have been shown to be able to bind and degrade chitin in fungal cell walls such as Cbp50 protein from *B. thuringiensis* (*Mehmood et al., 2011*), CHB2 protein from *S. reticuli* (*Kolbe et al., 1998*), CHB1 protein from *Streptomyces olivaceoviridis* (*Schnellmann et al., 1994*), CBP21 from *Serratia marcescens*, BtCBP protein in *B. thuringiensis*, and BliCBP in *Bacillus licheniformis* (*Manjeet et al., 2013*). Among them, Cbp50 protein has been shown to have antifungal activity. The ability of LPMO to bind chitin in the fungal cell wall is determined by the catalytic domain itself and in particular the CBMs, including CBM1, CBM2, CBM5, CBM5/12, CBM14 and CBM73 (*Forsberg & Courtade, 2023*; *Pan et al., 2023*).

As evidence to the mutualistic relationship, bacteria have been discovered associated with hyphae, spores, roots and fruiting bodies of fungi, inside fungal cells (endofungal bacteria) and on fungal surfaces (extrafungal bacteria) (*De Boer et al., 2005*; *Bonfante & Anca, 2009*). Recent research based on the co-culture method, *Embacher et al. (2021)* showed that in the brown rot fungus *Serpula lacrymans*, various bacteria coexist, mainly Gram-positive, and there are differences in the communities related to distinct mushroom parts. It was shown that Pseudomonadota ("Proteobacteria") bacteria do not form clear biofilm structures, but occur as independent cells scattered throughout and within tissues, sometimes even attached to fungal spores (*Embacher et al., 2023*). This proves that bacteria are able to adhere to fungal surfaces, but the adhesion mechanism has not yet been clarified.

In a recent publication, *Vaaje-Kolstad & Eijsink (2023)* while comparing the GbpA_2-bpA_3 complex in GbpA (*V. cholerae*) with the GbpA_2-Module-X complex of *Vh* LPMO10A (*Vibrio harveyi or campbellii*) (*Zhou et al., 2023*) and CbpD (*Pseudomonas aeruginosa*) (*Dade et al., 2022*), they discovered structural similarities of the GbpA_2-GbpA_3 complex between these three proteins. Although the actual role of the GbpA_2-Module-X complex is unknown, nevertheless, considering the available functional data for GbpA, they supposed that proteins such as *Vh* LPMO10A (also?) have a role in the pathogenicity-related physiology of *Vibrio* species. Specifically, the combined GbpA_2 and Module-X mediate binding of the secreted GbpA protein to the surface of the *Vibrio* cells.

In our study, we discovered a group of LPMOs that have a multidomain structure similar to the GbpA protein of *V. cholerae* and were annotated to have chitin-specific activity. This group of LPMOs has functional domains structurally similar to the *V. cholerae* GbpA, including a catalytic domain and a CBM domain (*e.g.*, CBM5, CBM5/12 and CBM73, Fig. 4A) capable of binding to chitin, while the GbpA_2 and GbpA_3-like domains that are structurally similar to GbpA_2 and GbpA_3 of *V. cholerae* GbpA. Therefore, the question is what role these LPMOs play in the relationship between white rot fungi and bacterial communities. Whether LPMOs with combined GbpA_2 and GbpA_3 have similar properties to GbpA that mediate adhesion of host bacterial cells to fungal surfaces is an interesting hypothesis and target for further studies.

Interestingly, the 13 LPMOs containing the GbpA_2 and GbpA_3 domains were all produced by bacteria belonging to the class Gammaproteobacteria, phylum Proteobacteria (Table S3). Similarly, *V. cholerae* also belongs to the class Gammaproteobacteria. It is likely that the GbpA_2 - GbpA_3 complex is transmitted horizontally between LPMOs of bacteria of the class Gammaproteobacteria and that the existence of chitin-active GbpA-like LPMOs is not coincidental in the ecosystem between white-rot fungi and bacterial communities on decomposing trees in nature.

Another interesting thing is that bacterial humus samples collected around white-rot fungi in Cuc Phuong forest had a pH ranging from 6.9 to 7.3 (*Le et al., 2022*), while a pH of 7.0 was determined to be the pH that bacteria (*e.g.*, *Pseudomonas putida*) had the highest antagonism to the white-rot fungus (*Phanerochaete chrysosporium*) (*Radtke, Cook & Anderson, 1994*). Perhaps this is also consistent with the time humus samples were collected at the site of the white rot fungus in the regressing stage of its development.

Maybe that is the reason why the collected humus samples all contained bacteria that produced LPMOs mainly with chitin activity.

Finally, structure-based HMM profiles of known LPMOs from different families (AA9, AA10, AA11, and AA13) were applied to discover new LPMO families from genomes of ascomycetous fungi (*Voshol, Vijgenboom & Punt, 2017*). To our knowledge, this is the first time HMM profiles have been successfully used in LPMO mining from bacterial DNA metagenomic data.

A limitation of this study is that only two LPMOs of Group 1 and Subgroup 3.1 were expressed and tested for enzymatic activity. It would be better if all representative genes of the eight phylogenetic groups were expressed and enzyme activity assayed to demonstrate the LPMO activity of all groups. However, the results of sequence and structure analysis of these proteins allow us to think that they are LPMOs with chitin and cellulose activity.

## CONCLUSION

We analyzed the amino acid sequences and molecular structures of 31 putative LPMOs mined from metagenomic data of humus samples collected around white rot fungi in Cuc Phuong primary forest. These proteins have the amino acid sequence and spatial structure characteristic of cellulose- and chitin-active LPMOs. Structural analysis using Interpro and Alphafold2 revealed a group of multimodular LPMOs with functional domains similar to GbpA of *V. cholera*. Based on the structural characteristics of functional domains, it is possible to hypothesize the role of chitin-active GbpA-like LPMOs in the relationship between fungal and bacterial communities coexisting on decomposing trees in primary forests.

**Abbreviations**

| | |
|---|---|
| **CBM** | Carbohydrate Binding Module |
| **AAs** | Auxiliary Activities |
| **CBP** | Chitin Binding Protein |
| **HMM** | Hidden Markov Model |
| **CBM** | Carbohydrate Binding Module |
| **Fn3** | Fibronectin 3 |
| **Ig-like** | Immunoglobulin like |
| **PVDF** | PolyVinylidene DiFloride |
| **IPTG** | IsoPropyl $\beta$-D-1-ThioGalactopyranoside |
| **PCR** | Polymerase Chain Reaction |
| **MUSCLE** | MUltiple Sequence Comparison by Log-Expectation |
| **MEGA** | Molecular Evolutionary Genetics Analysis |
| **CAZy** | Carbohydrate-Active enZyme |
| **HPAEC-** | High Performance Anion-Exchange Chromatography with Pulsed |
| **PAD** | Amperometric Detection |
| **ICP-MS** | Inductively Coupled Plasma Mass Spectroscopy |

# ACKNOWLEDGEMENTS

We thank Professor Juergen Pleiss (Institute of Biochemistry and Technical Biochemistry, University of Stuttgart, Germany) for providing the facility for our PhD student T.K. Dao to annotate metagenomic DNA sequences using HMMER software.

### Funding

This research was supported by the Bilateral International Project MetagenLig (NDT.50.GER/18) of the Ministry of Science and Technology (MOST), Vietnam and the Research Support Project of the Vietnam Academy of Science and Technology (VAST) (NCVCC08.03/22-23) for Nam-Hai Truong. There was no additional external funding received for this study. The funders had no role in study design, data collection and analysis, decision to publish, or preparation of the manuscript.

### Grant Disclosures

The following grant information was disclosed by the authors:
Bilateral International Project MetagenLig: NDT.50.GER/18.
Research Support Project of the Vietnam Academy of Science and Technology (VAST): NCVCC08.03/22-23.

### Competing Interests

The authors declare there are no competing interests

### Author Contributions

- Nam-Hai Truong conceived and designed the experiments, analyzed the data, prepared figures and/or tables, authored or reviewed drafts of the article, and approved the final draft.
- Thi-Thu-Hong Le performed the experiments, analyzed the data, prepared figures and/or tables, and approved the final draft.
- Hong-Duong Nguyen performed the experiments, analyzed the data, prepared figures and/or tables, and approved the final draft.
- Hong-Thanh Nguyen analyzed the data, prepared figures and/or tables, and approved the final draft.
- Trong-Khoa Dao analyzed the data, prepared figures and/or tables, and approved the final draft.
- Thi-Minh-Nguyet Tran performed the experiments, analyzed the data, prepared figures and/or tables, and approved the final draft.
- Huyen-Linh Tran analyzed the data, prepared figures and/or tables, and approved the final draft.
- Dinh-Trong Nguyen analyzed the data, prepared figures and/or tables, and approved the final draft.

- Thi-Quy Nguyen performed the experiments, analyzed the data, prepared figures and/or tables, and approved the final draft.
- Thi-Hong-Thao Phan analyzed the data, prepared figures and/or tables, and approved the final draft.
- Thi-Huyen Do analyzed the data, prepared figures and/or tables, and approved the final draft.
- Ngoc-Han Phan analyzed the data, prepared figures and/or tables, and approved the final draft.
- Thi-Cam-Nhung Ngo analyzed the data, prepared figures and/or tables, and approved the final draft.
- Van-Van Vu analyzed the data, prepared figures and/or tables, authored or reviewed drafts of the article, and approved the final draft.

## Data Availability

The raw data is available in the Supplementary Files.

## Supplemental Information

Supplemental information for this article can be found online at http://dx.doi.org/10.7717/peerj.17553#supplemental-information.

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
