# Peer review of "Sequence and structure analyses of lytic polysaccharide monooxygenases mined from metagenomic DNA of humus samples around white-rot fungi in Cuc Phuong tropical forest, Vietnam"

_PeerJ, doi:10.7717/peerj.17553_

## Round 0.1 · original submission · Major Revisions

Dear Dr. Truong and colleagues:

Thanks for submitting your manuscript to PeerJ. I have now received two independent reviews of your work, and as you will see, the reviewers raised some concerns about the research and manuscript. Despite this, these reviewers are optimistic about your work and the potential impact it will have on research studying chitin metabolizing enzymes in metagenomic data. Thus, I encourage you to revise your manuscript, accordingly, considering all of the concerns raised by both reviewers.

Importantly, please ensure that an English expert has edited your revised manuscript for content and clarity. Please also ensure that your figures and tables contain all the information that is necessary to support your findings and observations, including complete legends.

Please update the list of references such that more modern works are cited. The references should also be one format and include those references deemed important and missing by the reviewers. These references may also improve the presentation of your findings within more relevant literature.

There are many minor suggestions by the reviewers that should greatly improve your manuscript.

Thus, I encourage you to revise your manuscript, accordingly, considering all the concerns raised by the three reviewers.

Good luck with your revision,

-joe

Reviewer 1 ·

Basic reporting

According to manuscript number #91757, this work, the authors describe the sequence and structure analysis of lytic polysaccharide monooxygenases from metagenomic library constructed from humus samples around white rot fungi in Cuc Phuong tropical forest, Vietnam, followed by function analysis of two selected LPMO from group 1 (GL0247266) and subgroup 3.1 (GL0183513). Overall, the paper is well-written and concise. However, the following points need to be addressed before the manuscript can be considered for publication.
1. There are several formats of references found in this manuscript. Therefore, please edit the reference format throughout the manuscript.

Experimental design

There are some specific suggestions related to experimental design as following:
1. Two LPMO sequences (GL0247266 and GL0185513) were selected for recombinant protein production. However, there are three protein codes GL0185513, GL0183513, and GL0183523 representing the same protein. For example, in Figure 6, GL0185513 was found in the legend, whereas GL0183513 was labeled in the figure. Therefore, please check and correct the protein code throughout the manuscript.
2. Regarding Ms/Ms analysis indicated that the purified GL0185513 contained smaller protein band of chaperone HtpG from E. coli (Figure 6), how the authors confirm whether that the chaperone HtpG did not interfere LPMO activity assay and β-chitin hydrolysis via HPAEC method. Where did chaperone HtpG come from? Why there is no chaperone HtpG detected in recombinant GL0247266.
3. Regarding β-chitin hydrolysis via HPAEC method (Figure 7), it would be beneficial for readers if the quantitative data of products obtained from hydrolysis reaction is added to represent the specificity and performance of two selected LPMOs on β-chitin hydrolysis.
4. Improve resolution of Figure 1-9 to be high resolution with more than 300 dpi.
5. Please delete OS= from Line 398 and edit to be Chaperone HtpG from Escherichia coli.
6. Regarding β-chitin hydrolysis in Figure 7, therefore please correct the sentence in Line 419-420 to be “The evidence demonstrated the synergy between LPMOs (GL0247266 and GL0185513) and chitinase in β-chitin depolymerization”.
7. It would be beneficial for the readers if the conserved motif in Figure 2 is presented according to group and subgroup of putative LPMOs, for example H-H-F motif is conserved in all putative LPMOs while tryptophan triad and some amino acids are conserved in some subgroup of either cellulose or chitin-specific LPMOs. In alternative way, the conserved motif in Figure 2 can be presented in sequence logo format to show the frequency of amino acids in each position.

Validity of the findings

Based on sequence and structure annotation of putative LPMOs in this study, the results are beneficial for LPMO classification and structure-function relationship elucidation of chitin-specific LPMOs. However, please add more information, for example experimental data and/or previous published data, to propose the possible role of chitin-active LPMOs in the relationship between fungi and bacteria in wood-decaying ecosystem (Line 601-643).

Reviewer 2 ·

Basic reporting

The text will require further work to become clear and consise.The manuscript is too long. For instance it contains long paragraphs listing predicted protein names and certain properties such phylogenetic subgroup.
The literature references are in general fine although not fully up to date.
The hypothesis behind this work is unclear to me.

Experimental design

I focused on the expression, purfication and activity determination of the two proteins. It must be more clearly stated if it is the predicted full-length, multi-domain protein that is expressed or just the predicted LPMO-domain. The authors should consult more recent literature on LPMO activity assays and modify the conditions. The combination of ascorbate (2 mM) and CuSO4 (10 uM) will react very fast and produce produce reactive oxygen species that are harmful to proteins. "Free" copper must be avoided in all LPMO assays and synergy assays with hydrolases.

Validity of the findings

I am not sure.

---

## Round 0.2 · accepted · Accept

Dear Dr. Truong and colleagues:

Thanks for revising your manuscript based on the concerns raised by the reviewers. I now believe that your manuscript is suitable for publication. Congratulations! I look forward to seeing this work in print, and I anticipate it being an important resource for groups studying chitin metabolizing enzymes in metagenomic data. Thanks again for choosing PeerJ to publish such important work.

Best,

-joe

Reviewer 1 ·

Basic reporting

Some reference bibliography needs to be corrected, for example species name in line 700 must be italic. Therefore, please edit the reference bibliography throughout the manuscript according to format of PeerJ.

Experimental design

The manuscript has been greatly improved. Importantly, the authors conducted more experiments to explain hypothesis and addressed point by point, and added more information and relevant discussion in the manuscript.

Validity of the findings

-

Additional comments

-

Reviewer 2 ·

Basic reporting

no comments

Experimental design

no comments

Validity of the findings

no comments

Additional comments

It is unclear to me in what way the author have tested the "the hypothesis of the role of chitin-active GbpA-like LPMO in the relationship between bacteria and white rot fungi". To simply show that these enzymes are encoded for by bacteria from an environmental sample does not seem to go very far in that direction.